# Optimizing delivery strategies for 3HP TB preventive treatment in Tanzania: A qualitative study on acceptability of family approach in HIV care and treatment centers

Doreen Pamba[1]*, Erica Sanga[2], Killian Mlalama[1], Lucas Maganga[1], Chacha Mangu[1], Anange Lwilla[1], Willyhelmina Olomi[1], Lilian Tina Minja[1], Issa Sabi[1], Riziki Kisonga[3], Emmanuel Matechi[3], Isaya Jelly[4], Peter Neema[3], Anath Rwebembera[4], Said Aboud[5], Nyanda Elias Ntinginya[1]

1 National Institute for Medical Research-Mbeya Medical Research Center, Mbeya, Tanzania, 2 National Institute for Medical Research-Mwanza Medical Research Center, Mwanza, Tanzania, 3 National Tuberculosis and Leprosy Programme, Ministry of Health, Dodoma, Tanzania, 4 National AIDS, STIs & Hepatitis Control Programme, Ministry of Health, Dodoma, Tanzania, 5 National Institute for Medical Research, Dar es Salaam, Tanzania

* dpamba@nimr-mmrc.org

**Data Availability Statement:** Study participants were assured of data confidentiality when securing informed consent. Some of the interview

## Abstract

Tanzania rolled-out a 12-dose, weekly regimen of isoniazid plus rifapentine (3HP) TB preventive treatment in January 2024. The 3HP completion rate is generally ≥80%, varying by delivery strategy and programmatic setting. Before the roll-out, a mixed methods study was conducted to assess whether a family approach involving family member support, SMS reminders, and three health education sessions was acceptable and optimized 3HP uptake and completion. This paper describes acceptability of the family approach among people living with HIV (PLHIV), treatment supporters (TS), and community health workers (CHWs). This was a qualitative descriptive study in 12 HIV care and treatment centers across six administrative regions. We purposively sampled 20 PLHIV, 12 CHWs for in-depth interviews, and 23 TS for three focus group discussions held between September and December 2023. The theoretical framework of acceptability guided thematic-content analysis using a framework approach. Participants understood that PLHIV have an increased risk for active TB and that 3HP provides shortened treatment for TB disease prevention. They learned about TB and 3HP through health education sessions, but participation of TS was low due to expensive transportation costs to clinics. Receiving support from a trusted person and SMS were perceived as good adherence reminders. The majority reported mild self-limiting side effects but expressed positive attitudes because of the shortened treatment, TB counseling, satisfaction from helping others, lifestyle and work alignment, and reduced work burden. Some PLHIV had difficulties identifying supportive family members, so they chose close friends or CHWs. The family approach to supporting 3HP adherence is widely accepted by PLHIVs, TS, and CHWs in the context of person-centered care that respects their preferences. We recommend its adoption in programmatic settings as a combined approach,

transcripts contain identifiable information thus, making raw data available will be a breach to confidentiality. The data that support the study findings are available from the corresponding author upon reasonable request. Please contact dpamba@nimr-mmrc.org for data availability requests.

**Funding:** This study was funded by the Global Fund through the National AIDS, STIs & Hepatitis Control Programme (NASHCoP), grant name TZA-H-MOF and grant number 1961. The funders had no role in study design, data collection and analysis, decision to publish, or preparation of the manuscript.

**Competing interests:** The authors have declared that no competing interests exist

considering changes made during the study. However, further research is warranted to assess its acceptance among other populations eligible for 3HP.

## Introduction

The co-infection of tuberculosis (TB) and human immunodeficiency virus (HIV) globally remains a public health problem and a leading cause of death among people living with HIV (PLHIV) [1]. Global TB-related deaths among PLHIV were estimated at 190,000 in 2021, a stabilizing trend since 2019 [2]. PLHIV are 18 times more likely to become infected with TB and develop active TB than the general population due to the weakened immune system [3]. In 2022, PLHIV co-infected with TB contributed to 6.3% of the 10.6 million global TB incident cases [4].

WHO recommends TB preventive treatment (TPT) for the prevention of active TB disease among PLHIV, household contacts of bacteriologically confirmed pulmonary TB persons, and clinical risk groups such as those receiving dialysis [4, 5]. The efficacy of the available range of TPT regimens is documented to be 60% to 90% [5]. The 2018 United Nations High-Level Meeting (UNHLM) on TB committed to expand global TPT coverage to 6 million PLHIV by 2022 [6]. Although 11.3 million PLHIV worldwide were reported to have received TPT in 2022 thus exceeding the 2018 UNHLM target of 6 million, a drop of 300,000 PLHIV on TPT was observed between 2021 and 2022 [4].

Tanzania used isoniazid preventive treatment (IPT) for the prevention of active TB disease among PLHIV since 2010 [7]. The IPT involves the use of self-administered isoniazid daily for 6 to 9 months [8, 9]. In 2022, IPT coverage and completion rates were reported to be 84% and 79%, higher than the national coverage and completion targets of 70%and 70% in the same year [10]. Countries are recommended by WHO to adopt short-course rifamycin-containing TPT regimens (1 to 4 months) that are evidenced to be equally effective, with less hepatotoxicity and higher completion rates compared to IPT [5, 11, 12]. The 3HP is among the recommended shorter TPT regimen that involves taking Isoniazid and rifapentine once weekly for 12 weeks. The WHO TPT recommendations of 2020 acknowledge the existence of limited evidence on 3HP self-administration (SAT) thus, calling upon additional research including those of digital technologies for adherence support [5]. This implies that evidence is needed for novel strategies optimizing the delivery of 3HP in routine HIV care settings.

Overall, many studies have documented 3HP completion to be ≥80%, higher than SAT IPT in varied TB burden countries under research or routine conditions, and have assessed varied 3HP delivery strategies such as; direct observed therapy (DOT), video directly observed therapy (VDOT), SAT alone, SAT with text reminders and facilitated SAT with few conditional clinic visits for DOT [11, 13–15]. However, 3HP completion rates vary depending on the type of delivery strategy and programmatic setting. A study conducted in the United States, Spain, Hong Kong, and South Africa showed decreasing treatment completion rates with 3HP DOT (87.2%), 3HP SAT with text reminders (76.4%) and 74.0% with 3HP SAT alone [16]. Nevertheless, 3HP SAT has better completion rate and is cost-effective compared to IPT [17].

The Tanzanian Ministry of Health prepared to introduce 3HP based on local feasibility and acceptability studies [18]. Other than DOT, there is limited evidence on optimal strategies for delivering 3HP that are feasible and acceptable in sub-Saharan African programmatic settings. We thus, planned to identify optimized 3HP delivery strategies to support informed decisions on appropriate adherence support strategies during programmatic adoption. We piloted a

package of interventions called the family approach to 3HP to determine acceptability, feasibility, and impact on uptake, adherence, and completion among PLHIV within the context of a sequential explanatory mixed methods research. As part of the mixed methods research, this paper reports the qualitative component that aimed to assess acceptability of the family approach to 3HP among community health workers (CHWs), PLHIV, and their treatment supporters (TS). The quantitative results will be published separately.

## The family approach to 3HP

This approach used three strategies to support PLHIV with adherence, completion, and adverse events monitoring of 3HP to improve the uptake of 3HP. They included; involving a family member identified by PLHIV, sending automated SMS to PLHIV and their TS, and administering three TPT health education sessions (enrolment, months 2 and 3). The SMS content was co-designed with healthcare workers (HCWs) and CHWs to prevent unintentional HIV status disclosure. A system was developed to automatically send SMS, with HCWs from respective HIV care and treatment centers (CTCs) logging in to customize the timing of automated messages based on PLHIV preferences. The SMS were sent daily for adverse event monitoring, weekly for medication adherence, and monthly for clinic visit reminders, but only to consenting PLHIV and their TS.

The strategies were studied in a prospective cohort within 12 randomly selected high-volume HIV CTCs of publicly owned health facilities across six purposively selected regions of high HIV prevalence in the country. The cohort formed a quantitative component of a sequential explanatory mixed methods design implemented between May and December 2023. The inclusion criteria were: PLHIV enrolled into CTCs on or after April 2018 to March 2023 who have identified TS willing to attend three health education sessions and those who have completed TB treatment but have not started TPT at the time of study recruitment. PLHIV were given 3HP weekly for 3 months, commencing on the 14th day post-HIV diagnosis. CHWs at the study CTCs were trained on 3HP to support HCWs in educating PLHIV, and some of them served as TS for PLHIV unable to identify a family member or trusted friend.

## Theoretical framework

The theoretical framework of acceptability (TFA) guides assessment of health intervention acceptability by using seven conceptually distinct constructs that capture dimensions of emotional and cognitive responses of those delivering and receiving interventions [19]. We defined the acceptability of the family approach as participants' willingness to use its three components together as a supportive strategy for 3HP adherence; therefore, we operationalized the TFA constructs to guide data collection, analysis, and interpretation of factors influencing acceptance and areas that may require redesign for improved acceptance (Table 1).

## Methods

### Study design

We used a qualitative descriptive study to understand perceptions of CHWs, PLHIV, and their TS regarding acceptability of using a family approach to 3HP.

### Study setting

The study was conducted in 12 CTCs randomly selected to initiate 3HP among PLHIV across 6 regions in the country namely; Dar es Salaam (Tegeta dispensary and Amana regional referral hospital), Mbeya (Mbalizi designated district hospital and Matundasi dispensary), Iringa

**Table 1. TFA constructs and operationalization in the study.**

| TFA construct | Definition | Application in the study |
|---|---|---|
| Intervention coherence | The extent to which the participant understands the intervention and how it works | Participants' understanding of TB, TPT, and the family approach to 3HP and importance of TB prevention |
| Self-efficacy | The participant's confidence that they can perform the behavior(s) required to participate in the intervention | Participants' confidence in using or supporting adherence to 3HP |
| Perceived effectiveness | The extent to which the intervention is perceived as likely to achieve its purpose | Perceived effectiveness of the family approach in improving 3HP adherence |
| Affective attitude | How an individual feels about the intervention | Feelings about using or supporting 3PH adherence |
| Ethicality | The extent to which the intervention has good fit with an individual's value system | Alignment with lifestyle and work responsibilities |
| Opportunity cost | The extent to which benefits, profits, or values must be given up to engage in the intervention | Efforts or challenges encountered in the implementation of the family approach |
| Burden | Perceived amount of effort that is required to participate in the intervention | |

(Ngome health center and Mafinga district hospital), Njombe (Njombe town council health center and St. Consolata hospital, Ikonda), Tabora (Tura dispensary and Igunga district hospital) and Mwanza (Misasi health center and Buhongwa dispensary). Permission to conduct the study within the CTC settings was sought from respective health facility authorities.

## Participants and sampling

Three groups of participants were purposively sampled: PLHIV receiving care, their TS, and CHWs in their respective CTCs. We recruited 20 PLHIV using 3HP for over a month, along with 23 TS of PLHIV recruited by HCWs based on their convenience. We maintained a list of CHWs supporting each CTC thus, 12 CHWs from each of the CTCs were recruited.

## Data collection

Data were collected from 14th September 2023 to 9th December 2023 by trained social scientists experienced in qualitative research. Semi-structured, in-depth interviews (IDIs) and focus group discussions (FGDs) were conducted in Swahili for an average of 40 minutes and 90 minutes respectively (see S1–S3 Files for complete guides), and were consented for audio-recording. IDIs were conducted with 20 PLHIV and 12 CHWs. Three FGDs of seven to eight participants were conducted with 23 TS of PLHIV enrolled in four CTCs. Discussion topics focused on the seven constructs of the TFA to inform interpretations of the acceptability of the family approach to 3HP. These were: knowledge of the family approach package, confidence in implementing respective roles of the intervention, perceptions regarding intervention effectiveness, attitudes towards the intervention, challenges encountered, and perceptions of intervention alignment with lifestyle and work responsibilities.

## Data analysis

Three of the six social scientists (DP, ES, and KM) engaged in data collection were also involved in data analysis. Audio-recordings of IDIs and FGDs were transcribed verbatim and translated into English by respective social scientists who collected the data and who were fluent in both English and Kiswahili. We adopted thematic-content analysis using a framework method to analyze the qualitative data. A sample of the translated transcripts from each of the participant groups were exchanged among the social scientists to verify the accuracy of verbatim transcriptions against the audio-recordings. This process enabled a deeper familiarity with

the data. Initial versions of codebooks and coding matrices for each participant group were developed by DP, based on research questions and the TFA constructs. These initial versions were then shared with ES and KM, who made revisions that were agreed upon as a team. Furthermore, the remaining transcripts were distributed to each of the social scientists for independent coding and charting. Themes were developed based on overarching meanings identified from groups of categories related to TFA constructs. The themes were presented in descriptions that included quotations to further illustrate the findings.

### Ethical considerations

Ethical approval was sought from the Mbeya Medical Research and Ethics Review Committee (MMREC) in Tanzania (*Ref. No. SZEC-2439/R.A/V.1/168*). Written consent was secured from eligible participants.

## Results

The findings are presented as five themes related to the TFA constructs. These are broadly categorized as factors promoting acceptance of the family approach to 3HP among a majority of participants, and those that need improvement for the approach to be adopted within programmatic settings.

### Demographic characteristics of study participants

Many PLHIV (8/20) were enrolled in care in CTCs from hospital levels. A great proportion had only primary level education (16/20) and many were earning income from farming (8/20) and small businesses (8/20). They had a median age of 38 years and were living with HIV for a median duration of 3.5 months. The majority of TS were female (14/23), and many were earning income from small businesses (9/23). Many of them supported PLHIV enrolled in care in CTCs from hospitals (13/23). As for the CHWs, many volunteered to support CTCs from hospital levels (10/12), and half of them (6/12) had primary-level education (See Table 2).

### Factors promoting acceptance of the family approach to 3HP

**Theme 1: Perceived importance of TB prevention.** This theme describes participants' understanding of TB, TPT, and the family approach to 3HP. Many PLHIV understood that TB is airborne and they were at increased risk because of low immunity caused by HIV, so they took 3HP once weekly for three months to prevent TB disease. They understood that the TS was closely involved in supporting 3HP adherence: *"They told me to choose my closest person. . .the close person I have here is my husband. . ."* (Female PLHIV, IDI, health center).

Similarly, many TS understood that untreated individuals transmit TB and that their clients were at risk for TB disease because of low immunity from HIV. They understood that the family approach intends to help the client at a family level, reminding them to take medications, identify and report 3HP side effects.

*"Sometimes, the patient may not adhere to the treatment. By reminding them, they can take their medication without hesitation or quitting. . ." (*Female TS, FGD, hospital*)*

CHWs understood that 3HP is a shortened TPT compared to INH which is given to someone at risk for TB but without symptoms. They also noted that it offers flexibility to accommodate client preferences:

**Table 2. Characteristics of PLHIV, CHWs, and TS.**

| Characteristic | | N (%) | N (%) | N (%) |
|---|---|---|---|---|
| | | **PLHIV** | **CHW** | **TS** |
| **Facility level** | | | | |
| Dispensary | | 6 (30) | 4 (33) | 10 (43.5) |
| Health Center | | 6 (30) | 3 (25) | - |
| Hospital | | 8 (40) | 5 (42) | 13 (56.5) |
| **Sex** | | | | |
| Male | | 10 (50) | 2 (17) | 9 (39.1) |
| Female | | 10 (50) | 10 (83) | 14 (60.9) |
| **Age** (Median, IQR) | | 38 years (32–46) | - | - |
| **Education level** | | | | |
| Primary | | 16 (80) | 6 (50) | 9 (39.1) |
| Secondary | | 1 (0.05) | 4 (33) | 11 (47.8) |
| College | | 1 (0.05) | 2 (17) | 3 (13.0) |
| None | | 2 (0.1) | - | - |
| **HIV duration** (Median, IQR) | | 3.5 months (3–6) | - | - |
| **CHW experience** (Median, IQR) | | | 7yrs (5–14) | |
| **Occupation** | | | | |
| Business person | | 8 (40) | - | 9 (39.1) |
| Farmer | | 8 (40) | - | 8 (34.8) |
| Boda boda Driver | | 1 (0.05) | - | |
| Construction | | 1 (0.05) | - | |
| Mining | | 1 (0.05) | - | |
| Mechanics | | 1 (0.05) | - | |
| Clerk | | | - | 1 (4.3) |
| Food vendor | | | - | 1 (4.3) |
| Tailor | | | - | 1 (4.3) |
| Mason | | | - | 2 (8.7) |
| **Profession** | | | | |
| Clinician | | | - | - |
| Nurse | | | - | 1 (4.3) |

"...*the client will choose the specific day to take them...he can choose Wednesday, so every Wednesday he will be taking them till three months"* (Female CHW, IDI, health center).

Additionally, they understood that the family approach involves a family member supporting TPT adherence. However, they reported that the TS could be any close confidant, not necessarily a family member.

**Theme 2: Self-confidence in 3HP adherence.** This theme describes the confidence of PLHIV in adhering to 3HP treatment and the motivations that bolstered the confidence of TS and CHWs in supporting them. PLHIV were confident in their ability to adhere to 3HP and clinic visits because they understood they were at increased TB risk.

*"I knew that I have already got this [HIV] disease therefore I have low immunity, so, I better take the [TB] preventive medicines..."* (Male PLHIV, IDI, dispensary)

TS reported strong motivation to support treatment often due to familial obligations or friendship. For instance, one TS expressed a sense of duty to help her mother maintain her

health, while another supported her sister to prevent psychological suffering. Few of the TS were motivated to continue supporting treatment because of having clients who independently remembered to take their medications.

*" The client has made me confident to continue supporting him well. . .he is not afraid of taking pills. . ."* (Male TS, FGD, dispensary)

One of the CHWs stated having empathy for PLHIV without close support thus, was motivated to assume the role of a TS.

*"It is when you see people losing their lives, they are suffering because of not having a helper or an adviser; that is when I was motivated that. . .I must encourage him so that he uses the medicine to save his life. . ."* (Female TS, FGD, hospital)

**Theme 3: Perceived effectiveness of family approach.** This theme describes participant perceptions of whether the family approach will increase uptake and adherence to 3HP. The PLHIV valued having a trusted individual as a TS to support adherence and the freedom to choose a TS based on their close relationships. They appreciated receiving varied advice and food support from their TS:

*". . .he [TS] even advises me on what I should eat, type of work I should do, sometimes he even sends the texts to remind me on different issues related with TPT"* (Male PLHIV, IDI, dispensary)

The automated SMS texts were perceived as good medication reminders since 3HP is a once-weekly dose. Additionally, some PLHIV felt cared for by the HCWs after receiving the SMS texts.

*". . .if the client takes the medication once a week, then it is easier to forget, but once he receives the TPT text then it is easier to remember or even if the TS receives the text, then it is easier for him to remind him. . ."* (Male CHW, IDI, dispensary)

PLHIV perceived that the health education sessions increased their knowledge of TB and TPT. One of them attributed the sessions as a supportive platform: "*. . .if you really pay attention to them, it helps you. . ."* (Male PLHIV, IDI, hospital). The TS stated that health education sessions did not only improve their client's health but also the community as a whole.

*". . .health education helps me. . .I will be able to support another person, be in the family or outside the family which is the general community"* (Female TS, FGD, health center)

**Theme 4: Positive attitudes towards family approach.** The theme describes participants' feelings and adjustments to their lifestyle or work due to using 3HP or supporting 3HP use in the context of the family approach. The PLHIV appreciated that 3HP is shorter than INH in preventing them from TB, others felt happy about having a TS and receiving TB and TPT counseling at the clinics.

*"When I heard that this medicine will protect you against acquiring TB, and I have already witnessed how TB clients are suffering, I got motivated and I was very happy. . ."* (Female PLHIV, IDI, dispensary).

TS perceived their role as a blessing to serve people's lives. One of them stated that even the clients consider them as their confidants and supporters to improve not only physical but also psycho-social well-being intended for the client's health and welfare. Participants generally reported no changes in lifestyle or work responsibilities due to using 3HP. CHWs liked 3HP because it is shorter, and using a TS simplified their work. They viewed the changes as not related to additional work but rather a change in the type of TPT education they had to provide.

*"...it reduces some of my burdens and makes my work effective because if I miss my client and he has no TS then it becomes very difficult for me to locate him, but if he has a TS...it is easier to reach the client through [him]..."* (Male CHW, IDI, hospital)

**Theme 5: Barriers to accepting the family approach.** The theme describes the efforts and challenges in implementing the family approach to 3HP, which can be targeted for improvement to gain wider acceptance of the family approach.

*Lack of a reliable familial support system.* Some PLHIV had difficulties in selecting a TS who is a family member because they feared disclosing their status to their relatives, lacked close friends, or did not see the need for a TS.

*"...I haven't selected any [TS] because I live alone...I have friends, but I do not have one who is close to whom I can tell my secret..."* (Male PLHIV, IDI, hospital)

*"...I just can't say that you need someone to supervise you so that you can take the medicine because this is your health and when you take the medicine, that means you are protecting yourself... "* (Female PLHIV, IDI, health center)

One of the TS stated that he did not have the time to converse with his client about his treatment progress due to being busy with work. A few others complained about conflicting work commitments:

*"...It requires me to be available all the time, due to our temporary jobs, we can't rest even during the weekend, so I ended up losing my job."* (Female TS, FGD, hospital)

*Limited financial resources.* Additional challenges encountered by TS included lack of transport fare for themselves or their clients, inability to support their clients' food costs, difficulties spending quality time with their clients, and issues with timely medication intake:

*"...because we are living a bit far from here [referring to the CTC]...therefore, from there [referring to their home] up to here and back sometimes is difficult because I lack money for transport"* (Male TS, FGD, dispensary)

*Negative attitudes towards second and third health education sessions.* Although TS reported being able to support their clients during the first health education sessions, one of them *"...felt like they were rushed. They had many customers to attend to and didn't have time to explain properly"* (Female TS, FGD dispensary). They stated that their clients often arrived late, thus causing clinic visit delays, and recommended conducting group sessions. Attendance to second and third sessions was perceived as unnecessary as they could get the information from their clients:

*". . .we were together, my client and me on the first day. . . your client will just go and bring explanations on what was said. . . you are called to come, and there is a long queue, it becomes difficult"* (Male TS, IDI, hospital)

Few TS mentioned the issue of self-stigma, where some clients did not accept being accompanied to the healthcare facility although they were positive with phone follow-up.

*". . .let us say it is about stigma. . . he said I cannot go with a person to the hospital because I will be seen that I am taken to the hospital. . .I have never attended with him. . . "* (Female TS, FGD, dispensary)

*Mobile network connectivity issues.* CHWs stated that some of the PLHIV were unwilling to receive SMS alerts and some who lived in remote areas were unable to receive them on time. They anticipated challenges that might arise when scaling up 3HP for programmatic use. These included difficulties in sending automated SMS reminders to clients who; do not own mobile phones, reside in rural areas with poor network coverage, have a tendency not to read text messages, or are unwilling to receive SMS alerts:

*"There are challenges because some of the clients at the beginning refused to receive these texts. . .he would say don't send them I will remember on my own. . .He is worried that someone else might see that text"* (Female CHW, IDI, dispensary)

## Discussion

This study assessed the acceptability of a family approach to delivering 3HP among PLHIV, TS, and CHWs. The findings show that the family approach, which consists of three components supporting 3HP adherence and completion, is widely accepted by PLHIV, their TS, and CHWs. Acceptance was influenced by: knowledge about the increased TB risk for PLHIV and the benefits of short-course TPT, valuing the shortened treatment duration and involvement of a trusted and motivated person, perceiving SMS alerts as good reminders of adherence, alignment with work and lifestyle, being confident in adhering to the weekly dose schedule, and flexibility in both TS selection criteria, and timing of automated SMS. Other studies in Africa have also noted that perceived TB risk facilitates 3HP acceptance [20, 21]. Based on these findings, incorporating the family approach that uses automated SMS as a 3HP treatment support strategy is likely to be well-accepted by PLHIV. Furthermore, the WHO advocates using digital technologies to attain the End TB Strategy 2035 targets [22] and research demonstrates that digital technologies enhance latent TB treatment adherence and are acceptable to PLHIV [13, 21, 23].

To improve wider acceptance across participant groups, it was necessary to make changes to the two components of TS and health education sessions based on concerns and challenges identified during the study. Some of the PLHIV found it difficult to choose a family member as their TS, and some preferred involving trusted friends or CHWs. This highlights the importance of flexibility in TS selection criteria. A study has shown that high 3HP completion rates are achievable in programmatic settings when delivery strategies are optimized to overcome barriers [24]; thus, the proposed flexibility in TS selection criteria will align with the current national HIV management guidelines that require PLHIV to identify a TS for antiretroviral therapy (ART) adherence support [25]. Therefore, the same person can support 3HP regardless of familial relations. Furthermore, although PLHIV and their TS perceived the health education sessions as opportunities to learn about TB and TPT, the low involvement of TS in the

second and third sessions suggests that conducting them at three-time intervals is unfeasible. Studies show that family involvement can effectively support compliance with TB prevention and treatment when family members have positive attitudes and sufficient training [26–28]. Further research is needed on other modalities that effectively engage TS for a comprehensive understanding of their role.

More importantly, we observed that a few PLHIV denied identifying a TS, receiving automated SMS, or attending health education sessions with their TS. The denial of a component of the family approach implies that the approach in combination is not a one-size-fits-all strategy and that the diverse needs of PLHIV have to be considered within person-centered care to improve adherence. The provision of patient-centered TPT services within differentiated service delivery models is called upon for improved TPT coverage and increased completion rates (29–32) and is emphasized as the first pillar of the WHO End TB strategy for achieving End TB targets [29]. These findings emphasize the importance of considering patient preferences in 3HP programs in high TB burden countries. PLHIV who are unable to receive treatment support from family members can opt for other trusted persons or receive SMS reminders.

Although the family approach is acceptable to participants, the qualitative design cannot quantify its effectiveness in increasing 3HP adherence and completion. Additionally, we cannot discriminate acceptance of the family approach between rural and urban settings; thus, further research is warranted. However, our findings add to the knowledge base of optimized delivery strategies for 3HP treatment support among PLHIV enrolled in CTCs. The following recommendations should be considered for programmatic adoption, either as isolated components or a combined approach, to enhance 3HP treatment adherence besides SAT: (i) implement flexibility in the choice of TS that allows for trusted individuals beyond family members. The TS should be able to support both ART and 3HP adherence. (ii) explore health education session delivery modalities other than facility-based to reinforce TB and 3HP knowledge, and (iii) incorporate an automated SMS system whose content maintains HIV status confidentiality and considers the preferences of PLHIV regarding the timing of text message reminders.

## Conclusions

The family approach to supporting 3HP adherence is widely accepted by PLHIVs, TS, and CHWs in the context of person-centered care that respects their preferences. We recommend its adoption in programmatic settings as a combined approach, taking into account changes made during the study. However, further research is warranted to assess its acceptance among other populations eligible for 3HP.

## Supporting information

**S1 File. In-depth interview guide for people living with HIV.**
(DOCX)

**S2 File. In-depth interview guide for community health workers.**
(DOCX)

**S3 File. Topic guide for treatment supporters.**
(DOCX)

## Acknowledgments

The authors are thankful to all the participants who consented to discuss the study. We thank Nasra Abdul, Simeon Mwanyonga, and Lonze Ndelwa for supporting data collection and

transcriptions. We commend the support from Dr Catherine Joachim on behalf of the Tanzania Ministry of Health for ensuring a supportive environment for the implementation of the study.

## Author Contributions

**Conceptualization:** Doreen Pamba, Erica Sanga, Chacha Mangu, Willyhelmina Olomi, Lilian Tina Minja, Issa Sabi, Isaya Jelly, Anath Rwebembera, Nyanda Elias Ntinginya.

**Data curation:** Doreen Pamba, Erica Sanga, Killian Mlalama.

**Formal analysis:** Doreen Pamba, Erica Sanga, Killian Mlalama.

**Project administration:** Doreen Pamba, Anange Lwilla, Willyhelmina Olomi, Nyanda Elias Ntinginya.

**Supervision:** Doreen Pamba.

**Visualization:** Lucas Maganga, Chacha Mangu, Anange Lwilla, Willyhelmina Olomi.

**Writing – original draft:** Doreen Pamba.

**Writing – review & editing:** Doreen Pamba, Erica Sanga, Killian Mlalama, Lucas Maganga, Chacha Mangu, Anange Lwilla, Willyhelmina Olomi, Lilian Tina Minja, Issa Sabi, Riziki Kisonga, Emmanuel Matechi, Isaya Jelly, Peter Neema, Anath Rwebembera, Said Aboud, Nyanda Elias Ntinginya.

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
