## [Decision Letter · Decision Letter 0]

4 Jun 2024

PGPH-D-24-00695

Optimizing delivery strategies for 3HP TB preventive treatment in Tanzania: A qualitative study on acceptability of family approach in HIV care and treatment centers

Dear Dr. Pamba,

Thank you for submitting your manuscript to PLOS Global Public Health. After careful consideration, we feel that it has merit but does not fully meet PLOS Global Public Health’s publication criteria as it currently stands. Therefore, we invite you to submit a revised version of the manuscript that addresses the points raised during the review process.

I would like to sincerely apologise for the delay you have incurred with your submission. It has been exceptionally difficult to secure reviewers to evaluate your study. We have now received two completed reviews; the comments are available below. The reviewers have raised significant scientific concerns about the study that need to be addressed in a revision.

Please revise the manuscript to address all the reviewer's comments in a point-by-point response in order to ensure it is meeting the journal's publication criteria. Please note that the revised manuscript will need to undergo further review, we thus cannot at this point anticipate the outcome of the evaluation process.

We look forward to receiving your revised manuscript.

Kind regards,

Miquel Vall-llosera Camps

Staff Editor

Journal Requirements:

Reviewers' comments:

Reviewer's Responses to Questions

**Comments to the Author**

1. Does this manuscript meet PLOS Global Public Health’s publication criteria? Is the manuscript technically sound, and do the data support the conclusions? The manuscript must describe methodologically and ethically rigorous research with conclusions that are appropriately drawn based on the data presented.

Reviewer #1: Yes

Reviewer #2: Yes

2. Has the statistical analysis been performed appropriately and rigorously?

Reviewer #1: N/A

Reviewer #2: Yes

3. Have the authors made all data underlying the findings in their manuscript fully available (please refer to the Data Availability Statement at the start of the manuscript PDF file)?

Reviewer #1: Yes

Reviewer #2: Yes

4. Is the manuscript presented in an intelligible fashion and written in standard English?

Reviewer #1: No

Reviewer #2: Yes

5. Review Comments to the Author

Reviewer #1: Dear co-authors, congratulations on a thorough manuscript addressing an important topic. Although I only have three comments (outlined below), I hope that you are able to use these to revise more generally for clarity and precision. the manuscript overall feels unnecessarily long, with many of the quotes not adding much value to the actual finding they represent.

1. The general writing requires editing. It is beyond the scope of a reviewer and the journal to do this. But it is needed. here are a few examples from just a short section. These occur throughout:

line 418 - 'The PLHIV viewed using a closed one'; what does that mean? A person who is close to the PLHIV? Close is not the same as closed. And a 'close one' still reads as quite odd.

lines - 418-437 - duplicated?

line 438 - commented, not commended?

2. The use of the Sekhon et al. framework on acceptability as a conceptual framework is appropriate. However, it's introduction to the reader (lines 171-188) lacks clarity. Further, the operationalization of the key concepts to your data requires additional explanation. It's good to use this to organize the findings in principle, but in practice the findings seem very labored in presentation. Perhaps the acceptability concepts could be reframed as questions of the data, used as sub-headings to the findings (e.g., Did the intervention 'make sense' (coherence) to participants?). These can then be answered more declaratively with "Yes", or "Yes, most of it, but not ... " and so on.

3. The discussion is somewhat speculative. How much more do we really learn about TPT support for PLHIV from these data? The novelty should be in the combining of support elements (treatment support, sms etc.) and the overall acceptability of this combined approach. Instead, much of the discussion is (a) about each element in isolation, and (b) quite equivocal about what was learned. The discussion and then conclusion can be made more punchy once these comments are addressed.

Reviewer #2: Thank you for the invitation to review this very interesting manuscript on a qualitative assessment of the acceptability of a family approach to the 3HP TPT in Tanzania. The manuscript is well-written and interesting.

Major concerns:

1) Can you describe the role of the community health workers (CHW’s) in this program? It is unclear from the description what role the CHW’s paly in prescribing, educating, sending SMS messages and supporting other aspects of the program.

2) Who is sending the SMS reminders? This is not clear in the description of the program.

3) The manuscript would be substantially strengthened by a summary set of recommendations on how the program should be re-designed based on this study. For example, should there be fewer than 3 training sessions? Should any other refinements be made?

Minor corrections:

1) Page 5 line 119 – ‘hypertoxicity’ I think is meant to be ‘hepatotoxicity’

2) Page 12, Table 1: There is no data in the row ‘CTC working experience (median, IQR)

6. PLOS authors have the option to publish the peer review history of their article (what does this mean?). If published, this will include your full peer review and any attached files.

**Do you want your identity to be public for this peer review?** For information about this choice, including consent withdrawal, please see our Privacy Policy.

Reviewer #1: **Yes: **Graeme Hoddinott

Reviewer #2: No

---

## [Decision Letter · Decision Letter 1]

11 Sep 2024

PGPH-D-24-00695R1

Optimizing delivery strategies for 3HP TB preventive treatment in Tanzania: A qualitative study on acceptability of family approach in HIV care and treatment centers

Dear Dr. Pamba,

Thank you for submitting your manuscript to PLOS Global Public Health. After careful consideration, we feel that it has merit but does not fully meet PLOS Global Public Health’s publication criteria as it currently stands. Therefore, we invite you to submit a revised version of the manuscript that addresses the points raised during the review process.

The reviewers have assessed your revised manuscript and their comments are available below. Reviewer 1 has requested some minor revisions including shortening some of the quotes and fixing typographic errors. Please review their comments and make the appropriate amendments to the manuscript.

We look forward to receiving your revised manuscript.

Kind regards,

Emma Campbell, Ph.D

Staff Editor

Journal Requirements:

Reviewers' comments:

Reviewer's Responses to Questions

**Comments to the Author**

1. If the authors have adequately addressed your comments raised in a previous round of review and you feel that this manuscript is now acceptable for publication, you may indicate that here to bypass the “Comments to the Author” section, enter your conflict of interest statement in the “Confidential to Editor” section, and submit your "Accept" recommendation.

Reviewer #1: All comments have been addressed

Reviewer #2: All comments have been addressed

2. Does this manuscript meet PLOS Global Public Health’s publication criteria? Is the manuscript technically sound, and do the data support the conclusions? The manuscript must describe methodologically and ethically rigorous research with conclusions that are appropriately drawn based on the data presented.

Reviewer #1: Yes

Reviewer #2: Yes

3. Has the statistical analysis been performed appropriately and rigorously?

Reviewer #1: N/A

Reviewer #2: Yes

4. Have the authors made all data underlying the findings in their manuscript fully available (please refer to the Data Availability Statement at the start of the manuscript PDF file)?

Reviewer #1: Yes

Reviewer #2: Yes

5. Is the manuscript presented in an intelligible fashion and written in standard English?

Reviewer #1: Yes

Reviewer #2: Yes

6. Review Comments to the Author

Reviewer #1: Thank you to the authors for addressing my previous comments. I still believe that the findings could be presented more succinctly, with shorter quotes, or perhaps having the quotes in a table. Also, there are still several typos throughout, e.g.: line 138 - sub-Saharan Africa, not Sub-Sahara Africa, or lines 172-174, the citation (19) should appear here, not after the next sentence, or line 199 surely 3HP TPT or just 3HP rather than 3HPTPT? But, overall, the manuscript is coherent and reports important data.

Reviewer #2: All comments addressed. Thank you.

7. PLOS authors have the option to publish the peer review history of their article (what does this mean?). If published, this will include your full peer review and any attached files.

**Do you want your identity to be public for this peer review?** For information about this choice, including consent withdrawal, please see our Privacy Policy.

Reviewer #1: No

Reviewer #2: No

---

## [Editor Report · Decision Letter 2]

18 Nov 2024

Optimizing delivery strategies for 3HP TB preventive treatment in Tanzania: A qualitative study on acceptability of family approach in HIV care and treatment centers

PGPH-D-24-00695R2

Dear Ms. Pamba,

We are pleased to inform you that your manuscript 'Optimizing delivery strategies for 3HP TB preventive treatment in Tanzania: A qualitative study on acceptability of family approach in HIV care and treatment centers' has been provisionally accepted for publication in PLOS Global Public Health.

Best regards,

Julia Robinson

Executive Editor